# Trends in *Pseudomonas aeruginosa* In Vitro Susceptibility to Ceftolozane/Tazobactam in Latin America: SMART Surveillance Program, 2016–2024

**DOI:** 10.3390/antibiotics14101018

**Published:** 2025-10-14

**Authors:** Mark G. Wise, James A. Karlowsky, Thales J. Polis, Fakhar Siddiqui, Katherine Young, Mary R. Motyl, Daniel F. Sahm

**Affiliations:** 1IHMA, 2122 Palmer Drive, Schaumburg, IL 60173, USA; jkarlowsky@sharedhealthmb.ca (J.A.K.); dsahm@ihma.com (D.F.S.); 2Department of Medical Microbiology and Infectious Diseases, Max Rady College of Medicine, University of Manitoba, Room 543-745 Bannatyne Avenue, Winnipeg, MB R3E 0J9, Canada; 3Global Medical & Scientific Affairs (GMSA), MSD Brazil, São Paulo 04583-110, Brazil; thales.jose.polis@merck.com; 4Merck & Co., Inc., Rahway, NJ 07065, USA; fakhar_siddiqui@merck.com (F.S.); katherine_young@merck.com (K.Y.); mary_motyl@merck.com (M.R.M.)

**Keywords:** SMART, surveillance, *Pseudomonas aeruginosa*, ceftolozane/tazobactam, Latin America

## Abstract

**Objectives**: To describe annual trends in the susceptibility of clinical isolates of *Pseudomonas aeruginosa* from Latin America to ceftolozane/tazobactam. **Methods**: The Study for Monitoring Antimicrobial Resistance Trends (SMART) surveillance program collected 10,188 *P. aeruginosa* isolates from 57 unique clinical sites in 12 Latin American countries from 2016 to 2024. MICs were determined by reference broth microdilution testing and interpreted using 2025 CLSI M100 breakpoints. **Results**: Overall, 86.3% of clinical isolates of *P. aeruginosa* collected in Latin America were susceptible to ceftolozane/tazobactam, including 45.5% of multidrug-resistant (MDR) isolates. From 2016 to 2024, annual percent susceptible values for ceftolozane/tazobactam ranged from 84.9% (2016, *n* = 779) to 89.2% (2023, *n* = 1144), with a statistically significant linear trend for increasing susceptibility (*p* = 0.024; Cochran–Armitage test for trend). However, limiting analysis solely to the 14 clinical sites, from six countries, that participated in each of the nine years (*n* = 4565) indicated that the annual percent susceptible values for ceftolozane/tazobactam remained unchanged from 2016 (82.6%) to 2024 (83.9%) (*p* = 0.367; percent susceptible value range, 82.6 to 89.1%). Every year, from 2016 to 2024, all *P. aeruginosa* isolates from pediatric patients (<18 years of age) were consistently more susceptible to ceftolozane/tazobactam than those from adult patients (90.3 to 95.0%/year versus 83.3 to 88.6%/year, respectively). Significant variation (*p* < 0.05) in annual ceftolozane/tazobactam percent susceptible values was not observed for isolates from blood, intra-abdominal, and respiratory tract sources, while isolates from urine showed a trend of increasing ceftolozane/tazobactam susceptibility from 73.1% (2018, *n* = 145) to 90.6% (2023, *n* = 117) (*p* < 0.0001). Among individual countries that participated each year, *P. aeruginosa* isolates from all except Guatemala displayed stable or increasing rates of susceptibility to ceftolozane/tazobactam. **Conclusions**: Since it was first tested by the SMART program in 2016, and for 8 years thereafter, the in vitro activity of ceftolozane/tazobactam has remained consistent against clinical isolates of *P. aeruginosa* from the Latin American region (overall, 86.3% susceptible), with limited resistance development restricted to specific clinical sites.

## 1. Introduction

*Pseudomonas aeruginosa* is a leading cause of healthcare-associated infections, particularly in patients with compromised immune systems, such as those in intensive care units, with burns, or suffering from chronic respiratory conditions like cystic fibrosis [1]. This organism possesses intrinsic antimicrobial resistance mechanisms, including efflux pumps, low outer membrane permeability, and a chromosomal AmpC-type β-lactamase, as well as the ability to acquire additional resistance determinants and develop chromosomal mutations that can lead to antimicrobial treatment failure [2]. In Latin America, the prevalence of multidrug-resistant (MDR) *P. aeruginosa* isolates has become a major clinical and public health concern, complicating empirical treatment and resulting in increased patient morbidity and mortality [3].

Ceftolozane was developed to address the growing resistance of *P. aeruginosa* to marketed β-lactams and was combined with tazobactam, an established β-lactamase inhibitor [4]. The combination ceftolozane/tazobactam has shown potent in vitro activity against MDR and carbapenem-resistant *P. aeruginosa* [5], including isolates with AmpC overexpression or efflux pump activity, although it is inactive versus strains harboring metallo-β-lactamases (MBLs) [6]. Since its approval for use in the United States in 2014, and European and Latin American countries in 2015, ceftolozane/tazobactam has become an important therapeutic option for treating complicated intra-abdominal infections, complicated urinary tract infections, and hospital-acquired pneumonia caused by *P. aeruginosa* [7].

Latin America has been recognized as a hotspot for high rates of resistance among Gram-negative bacilli, including *P. aeruginosa* [8]. The Study for Monitoring Antimicrobial Resistance Trends (SMART) global surveillance program is an ongoing international initiative that monitors trends in antimicrobial resistance among clinically important Gram-negative bacterial pathogens. The purpose of this study was to perform longitudinal analyses (i.e., investigate annual trends) of ceftolozane/tazobactam susceptibility among clinical isolates of *P. aeruginosa* collected in Latin American countries by SMART from 2016 to 2024, including comparisons of ceftolozane/tazobactam susceptibility for isolates from adult and pediatric patients, from different infection sources, and from different countries.

## 2. Results

From 2016 to 2024, Latin American clinical laboratories participating in the SMART global surveillance program collected a total of 65,527 isolates of Gram-negative bacilli, of which 10,188 (15.5%) were *P. aeruginosa*. The percent susceptible value for ceftolozane/tazobactam tested against all isolates of *P. aeruginosa* collected in Latin America from 2016 to 2024 was 86.3%, approximately 12–14 percentage points higher than for the antipseudomonal cephalosporins ceftazidime and cefepime and approximately 19 percentage points higher than for meropenem (Table 1). Overall, 23.1%, 23.0%, and 17.1% of the *P. aeruginosa* collected in Latin America were found to be resistant to piperacillin/tazobactam, ceftazidime, and cefepime, respectively. Ceftolozane/tazobactam inhibited 53.3% of piperacillin/tazobactam-resistant, 43.8% of ceftazidime-resistant, and 35.6% of cefepime-resistant isolates. In total, 26.6% of isolates were resistant to meropenem, and over half of these (53.2%) were susceptible to ceftolozane/tazobactam. In total, 2292 (22.5%) and 1143 (14.2%) isolates were identified as multidrug-resistant (MDR) and difficult-to-treat resistant (DTR), respectively. Against these more resistant isolate subsets, ceftolozane/tazobactam inhibited 45.5% of MDR and 31.9% of DTR phenotypes.

Trends in annual MDR, DTR, and ceftolozane/tazobactam-susceptible percentages over the 9-year period from 2016 to 2024 are shown in Figure 1. As considerable variability regarding country/site participation in the SMART program over this time period existed (see Appendix A), leading to potential bias, longitudinal analysis was performed twice—once including the totality of isolates collected and a second time including solely the isolates from the clinical sites that contributed each year. Evaluating all isolates (Figure 1A), the percent susceptible value for ceftolozane/tazobactam ranged from 84.9% in 2016 to 89.2% in 2023, with a marginal decrease in susceptibility in 2024 to 87.1%, demonstrating a statistically significant increasing trend in susceptibility over time (*p* = 0.024). MDR percentages ranged from 22.1% (2022) to 16.6% (2023), with no increasing or decreasing trend evident. DTR percentages were highest in 2016 (15.8%) and lowest in 2023 (12.2%), with a decreasing trend (*p* = 0.039). Evaluating isolates from the consistently contributing sites yielded similar results (Figure 1B); the percent susceptible value for ceftolozane/tazobactam ranged from 82.6% in 2016 to 89.1% in 2021, with 83.9% susceptible in 2024 and with no significant trend of increasing or decreasing susceptibility observed. MDR percentages were highest in 2016 (27.8%) and lowest in 2019, 2021, and 2023 (16.9% in each of those years), with no significant trend. In contrast, DTR percentages showed a decreasing trend (*p* = 0.009), ranging from 17.9% in 2016 to 10.1% in 2023. However, the MDR and DTR rates were higher, both in the full data set and the set from the continuously contributing sites, in 2024 compared to the previous year.

Although ceftolozane/tazobactam is approved in the United States and Europe for treating complicated intra-abdominal and complicated urinary tract infections in pediatric patients, it is not yet licensed for pediatric use in some Latin American countries; thus, there is interest in ceftolozane/tazobactam activity against pathogens isolated from younger patients in the region. Figure 2 presents annual trends in ceftolozane/tazobactam susceptibility among *P. aeruginosa* from adult and pediatric (birth to <18 years old) patients, considering all isolates collected in Latin America from 2016 to 2024 (Figure 2A) and only those from the consistently contributing sites (Figure 2B). In both cases, isolates from pediatric patients were more susceptible to ceftolozane/tazobactam than those from adult patients each year. Among the totality collected each year, the difference in susceptibility between isolates from pediatric and adult patient populations ranged from 11.7% in 2016 to 4.2% in 2023. The pediatric/adult gap in susceptibility rates was even more pronounced among the isolates from the consistently contributing sites, ranging from a high of 16.7% in 2016 to 7.2% in 2021.

Figure 3 examines *P. aeruginosa* percent susceptible values for ceftolozane/tazobactam by infection source. Considering all isolates, the percent susceptible values for isolates from intra-abdominal and respiratory tract infections were consistently ≥83% each year, with no statistically significant trends (Figure 3A). Isolates from bloodstream infections were first collected for the SMART program in 2018, and 91.0% of the *P. aeruginosa* isolates that year were susceptible to ceftolozane/tazobactam; the percent susceptible values for bloodstream isolates decreased to 78.4% in 2020 but rose to 89.0% in 2024 (no significant trend noted). By contrast, urinary tract infection isolates demonstrated a strong linear trend of increasing susceptibility to ceftolozane/tazobactam, from 73.8% susceptible in 2016 to 88.1% susceptible in 2024 (*p* < 0.0001). Similar results were observed when considering the isolates from the consistently contributing sites (Figure 3B); no significant trends in ceftolozane/tazobactam susceptibility were observed for intra-abdominal, respiratory tract, or bloodstream infection isolates, while susceptibility among urinary tract infection isolates displayed a significant increasing trend (*p* = 0.011), from 70.0% susceptible in 2016 to 86.9% susceptible in 2024.

Among the countries that participated in SMART each year from 2016 to 2024, three (Argentina, Chile, and Panama) exhibited a significant trend of increasing susceptibility to ceftolozane/tazobactam (Figure 4A). No trend was observed for Colombia and Puerto Rico; however, the *P. aeruginosa* isolates collected in Guatemala displayed a significant trend of decreasing susceptibility to ceftolozane/tazobactam. Two hospitals in Guatemala participated in the SMART program—one contributing isolates each year from 2016 to 2024 and a second participating only from 2021 to 2024 (Appendix A). The decrease in *P. aeruginosa* susceptibility to ceftolozane/tazobactam in Guatemala was largely driven by the former, as evident in Figure 4B. The percent susceptible value for ceftolozane/tazobactam was ≥82.8% for the consistently participating site in Guatemala from 2016 to 2021 but exhibited a precipitous decrease in 2022 (72.2% susceptible), and, in 2024, just 25.5% of *P. aeruginosa* isolates collected by this site were susceptible to ceftolozane/tazobactam.

## 3. Discussion

Over the last decade in Latin America, ceftolozane/tazobactam has played a significant role as a treatment for infections caused by *P. aeruginosa*, a challenging, often MDR pathogen. During this time, the Latin American region has observed changes in susceptibility patterns associated with antibiotic pressure, emerging resistance mechanisms, and the introduction of new antimicrobial agents, including ceftolozane/tazobactam and ceftazidime/avibactam. An earlier study, conducted in Latin America prior to ceftolozane/tazobactam’s licensing, reported that, among *P. aeruginosa* collected from 2013 to 2015, 83.2% of isolates from Argentina, 90.6% of isolates from Brazil, 77.5% of isolates from Chile, and 90.5% of isolates from Mexico were ceftolozane/tazobactam-susceptible. The same study reported that 86.8% of all *P. aeruginosa* isolates tested were ceftolozane/tazobactam-susceptible [9], a value remarkably close to our finding of 86.3% susceptible in our combined 2016 to 2024 data set (Table 1). The Program to Assess Ceftolozane/Tazobactam Susceptibility (PACTS) global surveillance study tested 622 *P. aeruginosa* isolates from Latin American countries collected in 2015–2017 and reported a similarly high susceptibility rate to ceftolozane/tazobactam—90.8% [10]. On the other hand, in another study of 508 clinical isolates of *P. aeruginosa* collected from 2016 to 2017 at 20 hospitals in five Latin American countries, just 68.1% were ceftolozane/tazobactam-susceptible. This included percent susceptible values of 80.6% for isolates from Chile, 70% for isolates from Argentina, 68.3% for isolates from Brazil, 66.1% for isolates from Colombia, and 64.4% for isolates from Mexico [11]. While the reported value for isolates from Chile is consistent with our finding in the present study, the precent susceptible values for Argentina, Colombia, and Mexico were lower than we observed in each year of the SMART program. Tuon et al. examined a set of 132 *P. aeruginosa* isolates collected in Brazil in 2016–2017 and reported that 84.9% of isolates were ceftolozane/tazobactam-susceptible [12]. Brazil did not participate in the SMART program in 2024; however, from 2016 to 2023, 92.0% of the 1400 *P. aeruginosa* collected for the program were ceftolozane/tazobactam-susceptible.

In the current study, country-specific data revealed that, in most Latin American countries, *P. aeruginosa* susceptibility to ceftolozane/tazobactam remained highly stable from 2016 to 2024 when only clinical sites that participated each year were considered, with the notable exception of one hospital in Guatemala (Figure 4B). All *P. aeruginosa* isolates collected by this clinical site in 2016, 2017, and 2019 were ceftolozane/tazobactam-susceptible; however, in 2023, only 53.8% of submitted isolates were ceftolozane/tazobactam-susceptible, and, in 2024, only 25.5% of submitted isolates were ceftolozane/tazobactam-susceptible. Although the molecular characterization of ceftolozane/tazobactam-non-susceptible isolates is beyond the scope of our current report, we did note that, in 2023, among the 22 meropenem-non-susceptible isolates that were examined for β-lactamase carriage from this clinical site, 18 harbored an MBL (nine carried IMP-1 and nine carried VIM-2) (unpublished internal data on file). Previous work by the Antimicrobial Testing Leadership and Surveillance (ATLAS) global surveillance program reported that, among meropenem-non-susceptible *P. aeruginosa* collected in Guatemala from 2017 to 2019, approximately 38% (*n* = 42) carried a VIM MBL [13]. Despite regulatory reform in Guatemala, antibiotics remain widely available in convenience stores and are frequently dispensed without prescription [14]. This challenge to antibiotic stewardship in the country may contribute to increased resistance rates.

Limitations of the present study include its narrow scope (one antimicrobial agent against one bacterial species) and the fact that there were changes in participation by individual medical centers and countries over the years surveyed. In fact, all in vitro antimicrobial susceptibility testing surveillance programs can be challenged by varying participation among individual medical centers and countries when evaluating annual trends. For the 9-year trends presented here, we would have had to exclude the majority of sites (>75%) and isolates (>55% of the total collected) if we had focused solely on continuously participating sites. Given this data limitation, we opted to present trends for both the continuously participating clinical sites and all clinical sites. On a regional level, both sets of data lead to a similar conclusion—that is, that there is no evidence that the *P. aeruginosa* percent susceptible values for ceftolozane/tazobactam in Latin American countries changed significantly over time from 2016 to 2024, a time period over which the agent was used by care providers to treat their patients in these countries. Finer analysis, such as country-specific data, is more likely to show disparities, as can be seen in Figure 4. For example, in Chile, two clinical sites were consistent contributors to the program each year, while a third clinical site participated in all years except 2016. The inclusion of the third clinical site reduced the overall *P. aeruginosa* percent susceptible value for ceftolozane/tazobactam by >20 percentage points in 2017 and 2020 and by >30 percentage points in 2019. Clearly, outbreaks of β-lactam-resistant strains of *P. aeruginosa*, including MBL-carrying isolates, at a single institution can have a major impact on the overall percent susceptible values reported for a country when there are, overall, only a limited number of participating clinical sites.

## 4. Methods

### 4.1. Bacterial Isolates

Each clinical laboratory site that participated in the SMART global surveillance program was asked to collect consecutive, clinically significant isolates of aerobic or facultatively anaerobic Gram-negative bacilli from intra-abdominal infections (IAI), respiratory tract infections (RTI), urinary tract infections (UTI), and, starting in 2018, bloodstream infections (BSI). Isolates were restricted to one isolate per patient per Gram-negative species per year. All isolates were transported to a central laboratory (IHMA, Schaumburg, IL, USA), where they were re-identified using matrix-assisted laser desorption ionization time-of-flight (MALDI-TOF) mass spectrometry (Bruker Daltonics, Billerica, MA, USA) prior to antimicrobial susceptibility testing.

From 2016 to 2024, the SMART surveillance program collected a total of 10,188 *P. aeruginosa* isolates from 57 unique clinical sites in 12 countries in Latin America (Argentina, Brazil, Chile, Colombia, Dominican Republic, Ecuador, Guatemala, Mexico, Panama, Peru, Puerto Rico, and Venezuela); however, not all countries/sites participated each year. Appendix A provides the number of *P. aeruginosa* isolates collected over this time period by country and clinical site. In total, 14 clinical sites in 6 countries were consistent annual contributors to the program from 2016 to 2024.

### 4.2. Antimicrobial Susceptibility Testing

The Clinical and Laboratory Standards Institute (CLSI) reference broth microdilution method was used to determine isolate MICs [15]. MICs were interpreted as susceptible, intermediate, or resistant using 2025 CLSI M100 breakpoints [16]. *P. aeruginosa* isolates were classified as MDR based on resistance to ≥3 sentinel agents in differing drug classes (amikacin [aminoglycosides], aztreonam [monobactams], cefepime [cephalosporins], colistin [polymyxins], meropenem [carbapenems], levofloxacin [fluoroquinolones], and piperacillin/tazobactam [penicillin/β-lactamase inhibitor combinations]) [17]. Difficult-to-treat resistant (DTR) phenotypes were categorized using the criteria published by Kadri et al. [18]. Specifically, DTR phenotypes were defined by isolates not susceptible (intermediate or resistant) to all tested β-lactams (aztreonam, ceftazidime, cefepime, imipenem, meropenem, piperacillin-tazobactam), as well as fluoroquinolones (levofloxacin). The DTR definition excludes newer β-lactam/β-lactamase inhibitor combinations like ceftolozane/tazobactam.

### 4.3. Statistical Analysis

The Cochran–Armitage test was used to assess linear trends in percentage susceptible values from 2016 to 2024 using XLSTAT v2024.2.2 (Lumivero, Denver, CO, USA). A two-tailed *p*-value < 0.05 was considered statistically significant.

## 5. Conclusions

The current study demonstrates that, since 2016, when ceftolozane/tazobactam was added to the SMART global surveillance program, it has retained potent antimicrobial activity against *P. aeruginosa* in the Latin American region, and, to date, only localized decreases in percent susceptible values have been detected, likely the result of the spread of MBLs, which severely limit all therapeutic options for affected patients. However, the potential for increases in MDR and DTR isolates, and outbreaks with isolates carrying MBLs and other mechanisms of β-lactam resistance, warrants continued monitoring, both locally and by international surveillance programs.

## Figures and Tables

**Figure 1 antibiotics-14-01018-f001:**
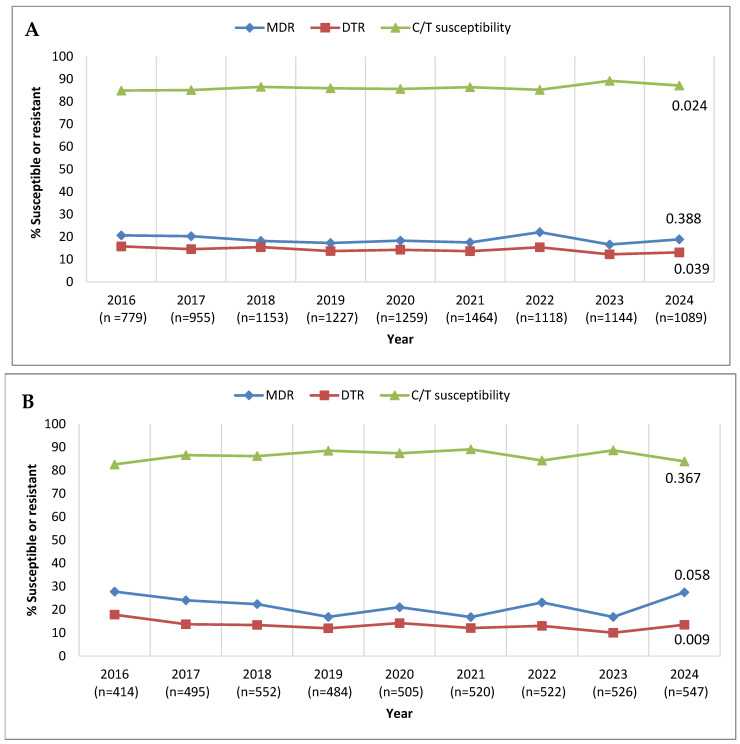
Longitudinal trends from 2016 to 2024 in the percentage of *P. aeruginosa* isolates identified as multidrug-resistant (MDR), difficult-to-treat resistant (DTR), and ceftolozane/tazobactam-susceptible (C/T-susceptible) among (**A**) all isolates collected in Latin America and (**B**) isolates collected from clinical sites that participated in the SMART program each year from 2016 to 2024. Significance in trends over time was determined by the Cochran–Armitage test: two-tailed *p*-values are shown in the figure (*p* < 0.05 was considered statistically significant). Percentages presented in this figure are provided in Appendix A.

**Figure 2 antibiotics-14-01018-f002:**
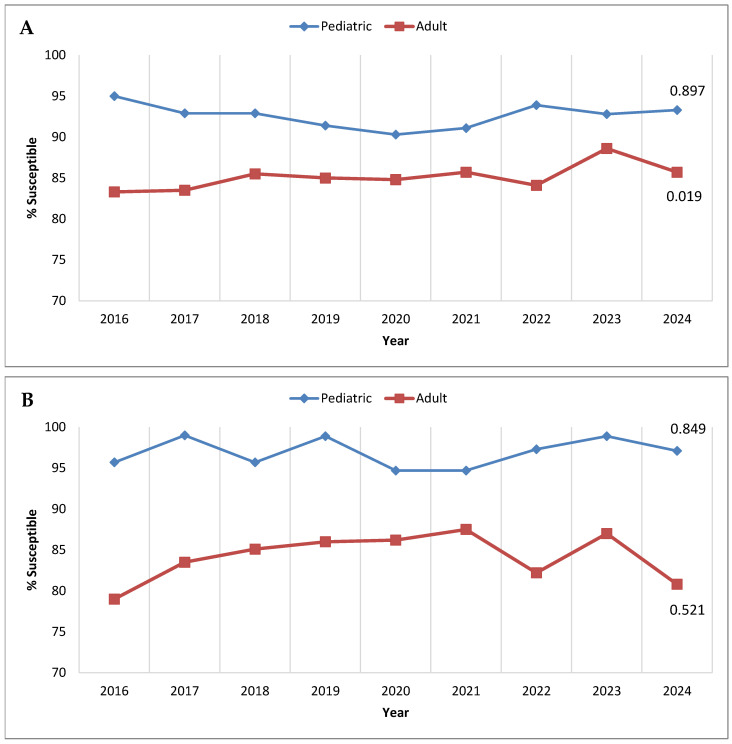
Longitudinal trends from 2016 to 2024 in the percentage of *P. aeruginosa* isolates testing as ceftolozane/tazobactam-susceptible, stratified by patient age, among (**A**) all isolates collected in Latin America and (**B**) isolates collected from clinical sites that participated in the SMART program each year from 2016 to 2024. Significance in trends over time was determined by the Cochran–Armitage test: two-tailed *p*-values are shown in the figure (*p* < 0.05 was considered statistically significant). Percentages presented in this figure are provided in Appendix A.

**Figure 3 antibiotics-14-01018-f003:**
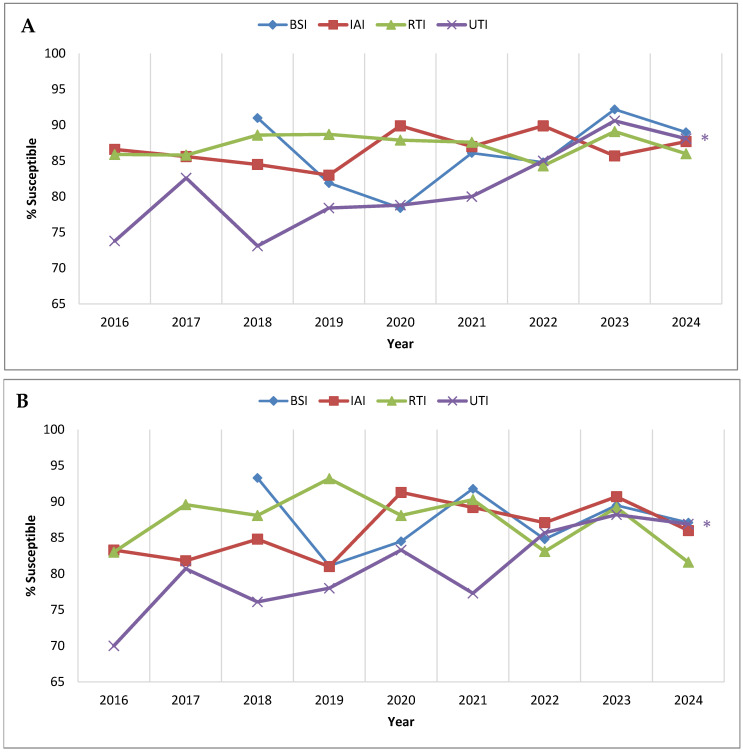
Longitudinal trends from 2016 to 2024 in the percentage of *P. aeruginosa* isolates testing as ceftolozane/tazobactam-susceptible, stratified by infection source, among (**A**) all isolates collected in Latin America and (**B**) isolates collected from clinical sites that participated in the SMART program each year from 2016 to 2024. Significance in trends over time was determined by the Cochran–Armitage test: two-tailed *p*-values are shown in the figure (*p* < 0.05 was considered statistically significant). * indicates increasing trend (*p* < 0.05). Percentages, numbers of isolates, and *p*-values presented in this figure are provided in Appendix A. Abbreviations: BSI, bloodstream infection; IAI, intra-abdominal infection; RTI, respiratory tract infection; UTI, urinary tract infection. BSI isolates were not tested in 2016 and 2017.

**Figure 4 antibiotics-14-01018-f004:**
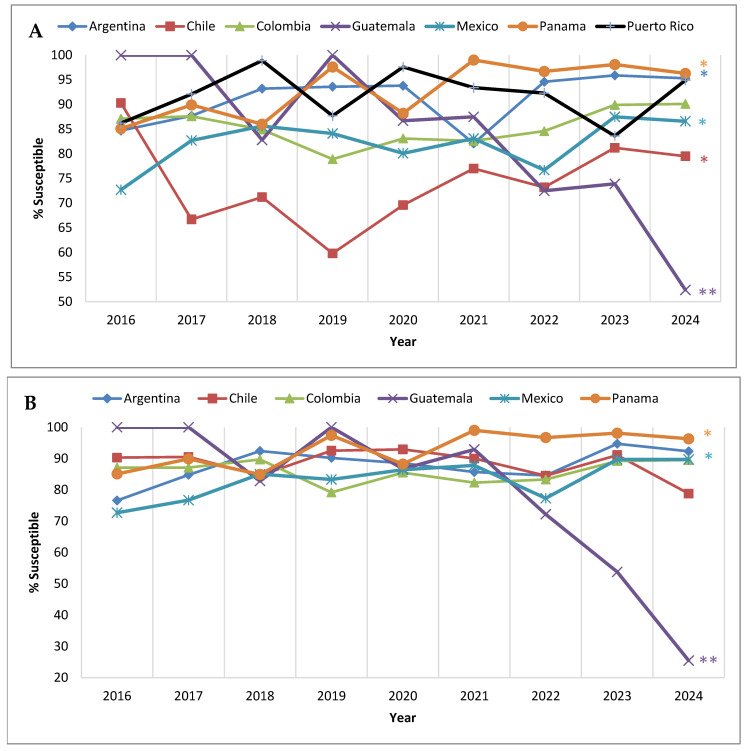
Longitudinal trends from 2016 to 2024 in the percentage of *P. aeruginosa* isolates that tested as ceftolozane/tazobactam-susceptible, by country, among (**A**) all isolates collected in the Latin American region from countries that participated in the SMART program each year and (**B**) isolates collected from clinical sites that participated each year from 2016 to 2024. Significance in trends over time determined by Cochran–Armitage test. * indicates significant increasing trend and ** indicates significant decreasing trend (*p* < 0.05). Percentages, numbers of isolates, and *p*-values presented in this figure are provided in Appendix A.

**Table 1 antibiotics-14-01018-t001:** Percent susceptible values for ceftolozane/tazobactam and comparator agents against clinical isolates of *P. aeruginosa* collected in Latin American countries from 2016 to 2024.

		% Susceptible								
Phenotype	*n*	C/T	IPM	MEM	CAZ	FEP	TZP	ATM	LVX	AMK
All isolates	10,188	86.3	58.1	67.2	72.8	74.1	69.1	64.2	64.4	85.8
CAZ-resistant	2341	43.8	19.2	21.3	0	5.5	3.5	11.6	23.4	51.1
FEP-resistant	1745	35.6	12.8	12.8	4.2	0	3.3	11.7	14.4	44.2
TZP-resistant	2358	53.3	21.2	23.6	7.0	10.4	0	11.7	24.9	55.0
MEM-resistant	2715	53.2	1.7	0	29.5	28.8	20.3	18.3	16.8	55.2
MDR	2292	45.5	11.8	11.2	11.6	8.6	4.4	10.7	11.2	45.7
DTR	1443	31.9	0	0	0	0	0	0	0	38.0

Abbreviations: C/T, ceftolozane/tazobactam; IPM, imipenem; MEM, meropenem; CAZ, ceftazidime; FEP, cefepime; TZP, piperacillin/tazobactam; ATM, aztreonam; LVX, levofloxacin; AMK, amikacin; MDR, multidrug-resistant; DTR, difficult-to-treat resistant.

## Data Availability

Data presented in this manuscript are available from the corresponding author upon reasonable request.

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
