# Peer review of "Trends in Pseudomonas aeruginosa In Vitro Susceptibility to Ceftolozane/Tazobactam in Latin America: SMART Surveillance Program, 2016–2024"

_antibiotics, 2025, doi:10.3390/antibiotics14101018_

Round 1
Reviewer 1 Report
Comments and Suggestions for Authors
The manuscript (DI: antibiotics-3906663) presents a multicenter surveillance study describing trends on susceptibility of Pseudomonas aeruginosa to ceftolozane/tazobactam across Latin America using the SMART program database (2016–2024). The study provide regional antimicrobial resistance data.
Below are my comments that need to be addressed for clarity and consistency.
- The objective of the study is clear for a descriptive surveillance study. However, since the study is descriptive, please rephrase the objective more explicitly as ‘to describe annual trends’ rather than “to evaluate,” which could imply analytical or mechanistic work. Would make the scope of the study clearer to readership
- The objective is bit narrow (only one combo of drugs, only one pathogen). So authors may state this in limitations of the study.
- The numbers and percentages are somewhat confusing. For example, I didn’t get it in case of 2016: 84.9% (n=779), the “n” appears to represent the number of resistant isolates or what? So to improve clarity for readers, rephrase it in abstract and in results section too.
- In Figure 1A and 1B, the values are overlapping with the lines. It would be clearer if the labels were repositioned to avoid overlap.
- Line 132: I could not understand the term “constant sites.” Please explain more clearly.
- In method section line 279, Which method was used for antimicrobial susceptibility testing?
- Was ethics approval was needed? If so, write the approval number etc.
Author Response
Response to reviewers
Reviewer #1
Below are my comments that need to be addressed for clarity and consistency.
1. The objective of the study is clear for a descriptive surveillance study. However, since the study is descriptive, please rephrase the objective more explicitly as ‘to describe annual trends’ rather than “to evaluate,” which could imply analytical or mechanistic work. Would make the scope of the study clearer to readership
Response: We agree with the reviewer and have changed the first sentence in the abstract (line 15).
2. The objective is bit narrow (only one combo of drugs, only one pathogen). So authors may state this in limitations of the study.
Response: We agree with the reviewer and have added the limitation. We have re-written the beginning of the limitations paragraph (lines 241-246) to address this.
3. The numbers and percentages are somewhat confusing. For example, I didn’t get it in case of 2016: 84.9% (n=779), the “n” appears to represent the number of resistant isolates or what? So to improve clarity for readers, rephrase it in abstract and in results section too.
Response: This is typical notation for these types of studies. The “n” is commonly understood to stand for “number” of isolates. We feel this needs no further explanation and will be straightforward for the typical reader.
4. In Figure 1A and 1B, the values are overlapping with the lines. It would be clearer if the labels were repositioned to avoid overlap.
Response: We have moved the Cochran-Armitage Test p-values in the figure so that they’re more clear.
5. Line 132: I could not understand the term “constant sites.” Please explain more clearly.
Response: We have used the term “constant” sites to mean the same as “consistently-contributing” sites. Perhaps the latter term is more clear, so we have changed this line 132)
6. In method section line 279, Which method was used for antimicrobial susceptibility testing?
Response: The methodology for the antimicrobial susceptibility testing is fully described in Section 4.2. (CLSI broth microdilution)
7. Was ethics approval was needed? If so, write the approval number etc.
Response: Not required for this study.
Reviewer 2 Report
Comments and Suggestions for Authors
Reviewer comments:
The reviewed manuscript is entitled: “Trends in Pseudomonas aeruginosa In Vitro Susceptibility to Ceftolozane/Tazobactam in Latin America: SMART Surveillance Program, 2016–2024”.
The specific comments:
- In the manuscript, there is no mention of obtaining ethical clearance or ethical waiver before conducting the study using microbial strains. Provide a statement mentioning ethic clearance or ethical waiver.
- In Figure 2A, why do paediatric patients display a higher degree of drug susceptibility as compared to adults?
- Line 223-226 “In the current study, country-specific data revealed that in most Latin American countries, aeruginosa susceptibility to ceftolozane/tazobactam remained highly stable from 2016 to 2024 when only clinical sites that participated each year were considered, with the notable exception of one hospital in Guatemala (Figure 4B)”. Provide a probable explanation for this observation.
- The authors need to propose how the data generated in the current study can be applied to address the shortcomings associated with bacterial drug resistance. This information needs to be included in the Discussion (section 3).
- Mention the limitations of the study in the Discussion (section 3) e.g. sample size, sample site etc.
Author Response
Reviewer #2
The specific comments:
1. In the manuscript, there is no mention of obtaining ethical clearance or ethical waiver before conducting the study using microbial strains. Provide a statement mentioning ethic clearance or ethical waiver.
Response: Not required for this study. Note that we already have the following statements included:
Institutional review board statement: Not applicable. (line 313)
Institutional consent statement: Not applicable. (line 314)
2. In Figure 2A, why do paediatric patients display a higher degree of drug susceptibility as compared to adults?
Response: This is unknown. Adults typically receive many more courses of antibiotics over their lifetime than children. Perhaps in Latin America this cumulative exposure creates selective pressure that favors the survival and spread of resistant strains in adults. However, this is just speculation.
3. Line 223-226 “In the current study, country-specific data revealed that in most Latin American countries, P. aeruginosa susceptibility to ceftolozane/tazobactam remained highly stable from 2016 to 2024 when only clinical sites that participated each year were considered, with the notable exception of one hospital in Guatemala (Figure 4B)”. Provide a probable explanation for this observation.
Response: a possible explanation is provided on lines 237-240:
Despite regulatory reform in Guatemala, antibiotics remain widely available in convenience stores and are frequently dispensed without prescription [14]. This challenge to antibiotic stewardship in the country may contribute to increased resistance rates.
4. The authors need to propose how the data generated in the current study can be applied to address the shortcomings associated with bacterial drug resistance. This information needs to be included in the Discussion (section 3).
Response: The major conclusion of the study is the fact that ceftolozane/tazobactam activity against the pathogen, Pseudomonas aeruginosa, has not decreased appreciably over the 9-year time period examined. This has been reiterated in the new Conclusion section (lines 299-307).
5. Mention the limitations of the study in the Discussion (section 3) e.g. sample size, sample site etc.
Response: The limitations paragraph in the discussion (lines 241-261) mentions these concerns.